# Using hindsight to anchor past knowledge in continual learning

## Abstract

In continual learning, the learner faces a stream of data whose distribution changes over time. Modern neural networks are known to suffer under this setting, as they quickly forget previously acquired knowledge. To address such catastrophic forgetting, many continual learning methods implement different types of *experience replay*, re-learning on past data stored in a small buffer known as episodic memory. In this work, we complement experience replay with a meta-learning technique that we call "anchoring": the learner updates its knowledge on the current task, while keeping predictions on some *anchor points* of past tasks intact. These anchor points are learned using gradient-based optimization as to maximize forgetting of the current task, in *hindsight*, when the learner is fine-tuned on the episodic memory of past tasks. Experiments on several supervised learning benchmarks for continual learning demonstrate that our approach improves the standard experience replay in terms of both accuracy and forgetting metrics and for various sizes of episodic memories.

## 1 Introduction

This manuscript studies the problem of *continual learning*, where a machine learning model experiences a sequence of tasks. Each of these tasks is presented as a stream of input-output pairs, where each pair is drawn from the corresponding task probability distribution. Since the length of the learning experience is not specified a-priori, the learner can only assume a *single pass over the data*, and store nothing but a few examples into a fixed-size *episodic memory*. At all times during the lifetime of the model, predictions on examples from all tasks can be requested. Addressing continual learning is an important research problem, since it would waive the long-obsolete assumption of "identically and independently distributed data" impeding progress towards artificial intelligence, and allow the deployment of models learning in-the-wild. However, continual learning presents one major challenge, *catastrophic forgetting* (McCloskey & Cohen, 1989). That is, as the learner experiences new tasks, it quickly forgets previously acquired knowledge. This is a hindrance for state-of-the-art deep learning models, where all parameters are updated after observing each example.

Continual learning has received increasing attention from the scientific community during the last decade. The state of the art in algorithms for continual learning fall into three categories. First, *regularization-based* approaches reduce forgetting by restricting the updates in parameters that were important for previous tasks (Kirkpatrick et al., 2016; Aljundi et al., 2018; Chaudhry et al., 2018; Nguyen et al., 2018). However, when the number of tasks are large, the regularization of past tasks becomes obsolete, leading to representation drift (Titsias et al., 2019). Second, *modular approaches* (Rusu et al., 2016; Lee et al., 2017) add new modules to the learner as new tasks are learned. While modular architectures overcome forgetting by design, the memory complexity of these approaches scales with the number of tasks. Third, *memory-based* methods (Lopez-Paz & Ranzato, 2017; Hayes et al., 2018; Isele & Cosgun, 2018; Riemer et al., 2019; Chaudhry et al., 2019a), store a few examples from past tasks in a so-called episodic memory, to be revisited when training for a new task. Memory-based methods are the reigning state of the art, but remain a far-cry from the achievable performance by a simple multi-task learning baseline accessing all data at once. Despite intense research efforts, such gap in performance renders the problem of continual learning an open research question.

**Contribution** We propose Hindsight Anchor Learning (HAL), a continual learning approach to improve the performance of memory-based continual learning algorithms. HAL leverages meta-learning to regularize the training objective with one representational point per class per task, called *anchor*. Since it is desirable to preserve the performance of a learner during its lifetime, HAL constructs anchors as to maximize forgetting on the current task throughout the learning experience. We estimate the amount of forgetting that the learner would suffer on those anchors when learning future tasks in *hindsight*: that is, by measuring forgetting on a temporary predictor that has been fine-tuned on the episodic memories of past tasks. Then, the main parameter update of HAL is to minimize the loss on the currently observed mini-batch, while keeping the predictions at all anchors invariant.

**Results** We compare HAL to EWC (Kirkpatrick et al., 2016), AGEM (Chaudhry et al., 2019a), experience replay (Hayes et al., 2018; Riemer et al., 2019), and MER (Riemer et al., 2019), across four standard benchmarks in continual learning (MNIST permutations, MNIST rotations, split CIFAR-100, and split miniImageNet). In these experiments, HAL achieves state-of-the-art performance, improving accuracy by 5% and reducing forgetting by 20%. We show that these results hold for various sizes of episodic memories (between 1 and 5 examples per class per task). Finally, we perform an ablation study to show that learning anchors in hindsight is a critical factor to achieve state-of-the-art performance.

We now begin our exposition by reviewing the continual learning setup. The rest of the manuscript then presents our new algorithm HAL (Section 3), showcases its empirical performance (Section 4), surveys the related literature (Section 5), and offers some concluding remarks (Section 6).

## 2 CONTINUAL LEARNING SETUP

In continual learning, we experience a stream of data triplets $(x_i, y_i, t_i)$, each containing an input $x_i$, a target $y_i$, and a task identifier $t_i \in \mathcal{T} = \{1, \ldots, T\}$. Each input-target pair $(x_i, y_i) \in \mathcal{X} \times \mathcal{Y}_{t_i}$ is an identically and independently distributed example drawn from some unknown distribution $P(X_{t_i}, Y_{t_i})$, representing the $t_i$-th learning task. Without loss of generality, we assume that the tasks are experienced in order ($t_i \leq t_j$ for all $i \leq j$), and that the number of total tasks $T$ is not known a priori. Under this setup, our goal is to estimate a predictor $f = (w \circ \phi) : \mathcal{X} \times \mathcal{T} \to \mathcal{Y}$, composed by a featurizer $\phi : \mathcal{X} \to \mathcal{H}$ and a classifier $w : \mathcal{H} \to \mathcal{Y}$, that minimizes the multi-task error

$$\frac{1}{T} \sum_{t=1}^{T} \mathbb{E}_{(x,y) \sim P_t} \left[ \ell(f(x, t), y) \right], \tag{1}$$

where $\mathcal{Y} = \cup_{t \in \mathcal{T}} \mathcal{Y}_t$, and $\ell : \mathcal{Y} \times \mathcal{Y} \to \mathbb{R}$ is a suitable loss function.

In the sequel, and similarly to prior literature in continual learning (Lopez-Paz & Ranzato, 2017; Hayes et al., 2018; Riemer et al., 2019; Chaudhry et al., 2019a), we consider streams of data, which are *experienced only once*. Therefore, the learner cannot store or revisit any but a small amount of data triplets chosen to be stored in its episodic memory $\mathcal{M}$. More specifically, we consider tiny "ring" episodic memories, which contain the last $m$ observed examples per class for each of the experienced tasks, where $m \in \{1, 3, 5\}$. That is, considering as variables the number of experienced tasks $t$ and examples $n$, we study continual learning algorithms with a $O(t)$ memory footprint.

Following Lopez-Paz & Ranzato (2017), we monitor two statistics to evaluate the quality of continual learning algorithms: *final average accuracy*, and *final maximum forgetting*. First, the final average accuracy of a predictor is defined as

$$\text{Accuracy} = \frac{1}{T} \sum_{j=1}^{T} a_{T,j}, \tag{2}$$

where $a_{i,j}$ denotes the test accuracy on task $j$ after the model finished experiencing task $i$. That is, the final average accuracy measures the test performance of the predictor at each of the tasks after the continual learning experience has finished. Second, the final maximum forgetting is defined as

$$\text{Forgetting} = \frac{1}{T-1} \sum_{j=1}^{T-1} \max_{l \in \{1, \ldots, T-1\}} (a_{l,j} - a_{T,j}), \tag{3}$$

that is, the decrease in performance at each of the tasks between their peak accuracy and their accuracy after the continual learning experience has finished.

Finally, following Chaudhry et al. (2019a), we use the first $k < T$ tasks to cross-validate the hyper-parameters of each of the considered continual learning algorithms. These first $k$ tasks are not considered when computing the final average accuracy and maximum forgetting metrics.

## 3 HINSDIGHT ANCHOR LEARNING (HAL)

Our starting point to describe our proposed continual learning algorithm is the current state of the art methods based on experience replay (Hayes et al., 2018; Riemer et al., 2019; Chaudhry et al., 2019b). These are predictors $f_\theta$ that, during training, store a small amount of past observed triplets into an episodic memory $\mathcal{M} = \{(x', y', t')\}$. When facing a new mini-batch of observations $\mathcal{B} := \{(x, y, t)\}$ from task $t$, they employ the rule $\theta \leftarrow \theta - \alpha \cdot \nabla_\theta \, \ell(\mathcal{B} \cup \mathcal{B}_{\mathcal{M}})$ to update their parameters, where

$$\ell(\mathcal{A}) = \frac{1}{|\mathcal{A}|} \sum_{(x,y,t) \in \mathcal{A}} \ell(f_\theta(x, t), y) \tag{4}$$

denotes the average loss across a collection of triplets $\mathcal{A}$, and $\mathcal{B}_{\mathcal{M}}$ is a mini-batch constructed by sampling triplets *of previous tasks* from $\mathcal{M}$ at random. In general, $\mathcal{B}_{\mathcal{M}}$ is constructed to have the same size as $\mathcal{B}$, but can be smaller if the episodic memory $\mathcal{M}$ does not yet contain enough samples.

In the previous, the episodic memory $\mathcal{M}$ reminds the predictor about how to perform at past tasks using only a very small amount of datum. As such, the behaviour of the predictor outside the data stored in $\mathcal{M}$ is not guaranteed, and subject to worsen. Because of this reason, we would like to be more conservative, and propose to further fix the behaviour of the predictor at a collection of carefully constructed *anchor points* $e_{t'}$, one per class per past task $t'$, after a parameter update[1].

To implement this, we propose a two-step parameter update rule:

$$\tilde{\theta} \leftarrow \theta - \alpha \cdot \nabla_\theta \, \ell(\mathcal{B} \cup \mathcal{B}_{\mathcal{M}}),$$

$$\theta \leftarrow \theta - \alpha \cdot \nabla_\theta \left( \ell(\mathcal{B} \cup \mathcal{B}_{\mathcal{M}}) + \lambda \cdot \sum_{t' < t} \left( f_\theta(e_{t'}, t') - f_{\tilde{\theta}}(e_{t'}, t') \right)^2 \right). \tag{5}$$

The first step computes a temporary parameter vector $\tilde{\theta}$ by minimizing the loss at a minibatch from the current task $t$, and the episodic memory of past tasks (this is the usual experience replay parameter update). The second step employs nested optimization to perform the final update of the parameter $\theta$, which trades-off the minimization of $(a)$ the loss value at the current minibatch and the episodic memory, as well as $(b)$ the change in predictions at the anchor points for all past tasks. We use L2 over the unnormalized logits as a distance measure in Eq. 5, as we experimentally find it to be superior than cosine similarity or L1 distance. This two-step nested optimization bears a similarity to gradient-based meta-learning approach (Finn et al., 2017) where we restrict the inner update to only a single gradient step.

Next, let us discuss how to choose the anchor points. Recall that our objective is to preserve the performance of the current task throughout the entire continual learning experience. Ideally, we would like to choose each $e_t$ as a tool to maximally suppress the amount of forgetting about current task throughout the entire learning experience. For this, we are interested in letting $e_t$ be the example from task $t$ on which the loss would increase maximally had training with future tasks been performed. Such $e_t$ would capture instances where the current task would be forgotten the most in future training. Then, requiring the predictions to remain invariant at such $e_t$, by using Eq. 5, would minimize forgetting on the current task. Mathematically, the desirable $e_t$ and its label $y_t$ is given by:

$$(e_t, y_t) \leftarrow \underset{(x,y) \sim P_t}{\arg\max} \, \ell(f_{\theta_T}(x, t), y) - \ell(f_{\theta_t}(x, t), y), \tag{6}$$

where $\theta_t$ is the parameter vector after training on task $t$ and $\theta_T$ is the final parameter vector after the entire learning experience. Thus, keeping predictions intact on the pair $(e_t, y_t)$ above would

---

[1]To ease the exposition, the remaining of the section assumes only one anchor $e_{t'}$ per task $t'$. The extension to one anchor per class per task, as we use in our experiments, is straightforward.

maximally preserve the performance of task $t$. However, the idealistic Eq. 6 requires access to ($a$) the entire distribution $P_t$ as to compute the maximization, and ($b$) access to all future distributions $t' > t$ as to compute the final parameter vector $\theta_T$. Both are unrealistic assumptions.

To circumvent ($a$), we can recast Eq. 6 as to *learn* $e_t$ by initializing it at random and using $k$ gradient ascent updates for a given label $y_t$:

$$e_t \leftarrow e_t + \alpha \cdot \nabla_{e_t} \left( \underbrace{\ell(f_{\theta_T}(e_t, t), y_t) - \ell(f_{\theta_t}(e_t, t), y_t)}_{\text{Forgetting loss}} - \gamma \underbrace{(\phi(e_t) - \phi_t)^2}_{\text{Mean embedding loss}} \right), \qquad (7)$$

where we recall that $\phi$ denotes the feature extractor of the predictor, and $\phi_t$ is the neural mean embedding (Smola et al., 2007) of all observed examples from task $t$ which regularizes the constructed $e_t$ against the outliers. Since the feature extractor is updated after experiencing each data point, the mean embeddings $\phi_t$ are computed as running averages. That is, after observing a minibatch $\mathcal{B} = \{(x, y, t)\}$ of task $t$, we update:

$$\phi_t \leftarrow \beta \cdot \phi_t + (1 - \beta) \frac{1}{|\mathcal{B}|} \sum_{x \in \mathcal{B}} \phi(x), \qquad (8)$$

where $\phi_t$ is initialized to zero at the beginning of the learning experience. In our experiments, we learn one $e_t$ per class for each task. We fix the $y_t$ to the corresponding class label, and discard $\phi_t$ after training on task $t$. Learning $e_t$ in this manner circumvents the requirement of storing the entire distribution $P_t$ for a task.

Still, Eq. 7 requires the parameter vector $\theta_T$, to be obtained in the distant future after all learning tasks have been experienced. To waive this impossible requirement, we will *approximate the future by simulating the past*. That is, instead of measuring the forgetting that would happen after the model is trained at future tasks, we will measure the forgetting that happens when the model is fine-tuned at past tasks. In this way, we say that forgetting is estimated in *hindsight*, using past experiences. More concretely, after training on task $t$ and obtaining the parameter vector $\theta_t$, we minimize the loss during one epoch on the episodic memory $\mathcal{M}$ to obtain the temporary parameter vector $\theta_{\mathcal{M}}$, and recast Eq. 7 as:

$$e_t \leftarrow e_t + \alpha \cdot \nabla_{e_t} \left( \ell(f_{\theta_{\mathcal{M}}}(e_t, t), y_t) - \ell(f_{\theta_t}(e_t, t), y_t) - \gamma(\phi(e_t) - \phi_t)^2 \right). \qquad (9)$$

Our experiments show that the proposed approximation of future by replaying past data in hindsight reduces forgetting by 20% when compared to a standard experience replay baseline.

This completes the description of our proposed algorithm for continual learning, which combines experience replay with anchors learned in hindsight. We call our approach Hindsight Anchor Learning (HAL) and summarize the entire learning process as follows:

---

**Hindsight Anchor Learning (HAL)**

- Initialize $\theta \sim P(\theta)$ and $\{e_t \sim P(e)\}_{t=1}^T$ from normal distributions $P(\theta)$ and $P(e)$.
- Initialize $\mathcal{M} = \{\}$
- For each task $t = 1, \ldots, T$:
  - Initialize $\phi_t = 0$
  - For each minibatch $\mathcal{B}$ from task $t$:
    * Sample $\mathcal{B}_{\mathcal{M}}$ from $\mathcal{M}$
    * Update $\theta$ using Eq. 5
    * Update $\phi_t$ using Eq. 8
    * Update $\mathcal{M}$ by adding $\mathcal{B}$ in a first-in first-out (FIFO) ring buffer
  - Fine-tune on $\mathcal{M}$ to obtain $\theta_{\mathcal{M}}$
  - Build $e_t$ using Eq. 9 $k$ times
  - Discard $\phi_t$
- Return $\theta$.

---

## 4 EXPERIMENTS

We now evaluate the performance of HAL against a variety of state-of-the-art methods, across four popular continual learning benchmarks.

### 4.1 DATESETS AND TASKS

We perform experiments on four popular benchmarks for continual learning.

- *Permuted MNIST* is a variant of the MNIST dataset of handwritten digits (LeCun, 1998) where each task applies a fixed random pixel permutation to the original dataset. This benchmark contains 23 tasks, each with 1000 samples from 10 different classes.
- *Rotated MNIST* is another variant of MNIST, where each task applies a fixed random image rotation (between 0 and 180 degrees) to the original dataset. This benchmark contains 23 tasks, each with 1000 samples from 10 different classes.
- *Split CIFAR* is a variant of the CIFAR-100 dataset (Krizhevsky & Hinton, 2009; Zenke et al., 2017), where each task contains the data pertaining 5 random classes (without replacement) out of the total 100 classes. This benchmark contains 20 tasks, each with 250 samples per each of the 5 classes.
- *Split miniImageNet* is a variant of the ImageNet dataset (Russakovsky et al., 2015; Vinyals et al., 2016), containing a subset of images and classes from the original dataset. This benchmark contains 20 tasks, each with 250 samples per each of the 5 classes.

For all datasets, the first 3 tasks are used for hyper-parameter optimization (grids available in Appendix B). The learners can perform multiple epochs on these three initial tasks, that are later discarded for evaluation.

### 4.2 BASELINES

We compare our proposed model HAL to the following baselines.

- *Finetune* is a single model trained continually on the stream of data, without any regularization or episodic memory.
- *EWC* (Kirkpatrick et al., 2016) is a continual learning method that limits changes to parameters critical to past tasks, as measured by the Fisher information matrix.
- *AGEM* (Chaudhry et al., 2019a) is a continual learning method improving on (Lopez-Paz & Ranzato, 2017), which uses an episodic memory of parameter gradients to limit forgetting.
- *MER* (Riemer et al., 2019) is a continual learning method that combines episodic memories with meta-learning to limit forgetting.
- *ER-Ring* (Chaudhry et al., 2019b) is a continual learning method that uses a ring buffer as episodic memory.
- *Multitask* is an oracle baseline that has access to all data to optimize Eq. 1, useful to estimate an upper bound on the obtainable Accuracy (Eq. 2).
- *Clone-and-finetune* is an oracle baseline training one independent model per task, where the model for task $t'$ is initialized by cloning the parameters of the model for task $t' - 1$.

All baselines use the same neural network architectures: a perceptron with two hidden layers of 256 ReLU neurons in the MNIST experiments, and a ResNet18, with three times less feature maps across all layers, similar to Lopez-Paz & Ranzato (2017), their CIFAR and ImageNet experiments. The task identifiers are used to select the output head in the CIFAR and ImageNet experiments, while ignored in the MNIST experiments. Batch size is set to 10 for both the stream of data and episodic memories, across experiments and models. The size of episodic memories is set between 1 and 5 examples per class per task. The same unified code base runs all experiments, and is available at `https://bit.ly/2mw8bsE`.[2]

---

[2]The code base is anonymized and implements the Permuted MNIST experiments.

Table 1: Accuracy (Eq. 2) and Forgetting (Eq. 3) results of continual learning experiments. Averages and standard deviations are computed over five random seeds. When used, episodic memories contain up to one example per class per task. Last two rows are oracle baselines.

| Method | Permuted MNIST | | Rotated MNIST | |
|---|---|---|---|---|
| | Accuracy | Forgetting | Accuracy | Forgetting |
| Finetune | 53.5 (±1.46) | 0.29 (±0.01) | 41.9 (±1.37) | 0.50 (±0.01) |
| EWC | 63.1 (±1.40) | 0.18 (±0.01) | 44.1 (±0.99) | 0.47 (±0.01) |
| AGEM | 62.1 (±1.39) | 0.21 (±0.01) | 50.9 (±0.92) | 0.40 (±0.01) |
| MER | 69.9 (±0.40) | 0.14 (±0.01) | 66.0 (±2.04) | 0.23 (±0.01) |
| ER-Ring | 70.2 (±0.56) | 0.12 (±0.01) | 65.9 (±0.41) | 0.24 (±0.01) |
| **HAL (ours)** | **73.6** (±0.31) | **0.09** (±0.01) | **68.4** (±0.72) | **0.21** (±0.01) |
| Clone-and-finetune | 81.4 (±0.35) | 0.0 | 87.5 (±0.11) | 0.0 |
| Multitask | 83.0 | 0.0 | 83.3 | 0.0 |

| Method | Split CIFAR | | Split miniImageNet | |
|---|---|---|---|---|
| | Accuracy | Forgetting | Accuracy | Forgetting |
| Finetune | 42.9 (±2.07) | 0.25 (±0.03) | 34.7 (±2.69) | 0.26 (±0.03) |
| EWC | 42.4 (±3.02) | 0.26 (±0.02) | 37.7 (±3.29) | 0.21 (±0.03) |
| AGEM | 54.9 (±2.92) | 0.14 (±0.03) | 48.2 (±2.49) | 0.13 (±0.02) |
| MER | 49.7 (±2.97) | 0.19 (±0.03) | 45.5 (±1.49) | 0.15 (±0.01) |
| ER-Ring | 56.2 (±1.93) | 0.13 (±0.01) | 49.0 (±2.61) | 0.12 (±0.02) |
| **HAL (ours)** | **60.4** (±0.54) | **0.10** (±0.01) | **51.6** (±2.02) | **0.10** (±0.01) |
| Clone-and-finetune | 60.3 (±0.55) | 0.0 | 50.3 (±1.00) | 0.0 |
| Multitask | 68.3 | 0.0 | 63.5 | 0.0 |

## 4.3 RESULTS

Table 1 summarizes the main results of our experiments. First, our proposed HAL is the method achieving maximum Accuracy (Eq. 2) and minimal Forgetting (Eq. 3) at all benchmarks. This does not include oracle baselines *Multitask* (which has access to all data simultaneously) and *Clone-and-finetune* (which trains a separate model per task). Second, the relative gains from the second-best method *ER-Ring* to HAL are significant, confirming that the anchoring objective (Eq. 5) allows experience-replay methods to generalize better from the same amount of episodic memory.

Figure 1 shows a more fine grained analysis of average accuracy as new tasks are learned on Permuted MNIST and Split CIFAR. HAL preserves the performance of a predictor more effectively than other baselines.

Table 2 shows the Accuracy of methods that employ an episodic memory, when we allow the size of this memory to increase; 3 or 5 examples per class per task, resulting in a total memory size of 600 or 1000 for MNIST experiments, and 255 or 425 for CIFAR and ImageNet experiments. The corresponding numbers for Forgetting are given in Appendix Table 4. HAL outperforms all competitors at all benchmarks.

Figure 2 provides the training time of all the continual learning baselines on MNIST benchmarks. Although HAL adds a overhead on top of experience replay baseline, it is significantly faster than, MER, another meta-learning approach to reduce forgetting.

Finally, Table 3 summarizes the results of an ablation study to better understand the impact of anchor selection. We compare selecting anchors as random noise (Random-Anchors), as a random example for the associated class and task (Data-Anchors), or our proposed optimization in hindsight procedure (HAL). Our results confirm that optimizing anchors in hindsight is the most effective strategy in terms of both accuracy and forgetting metrics.

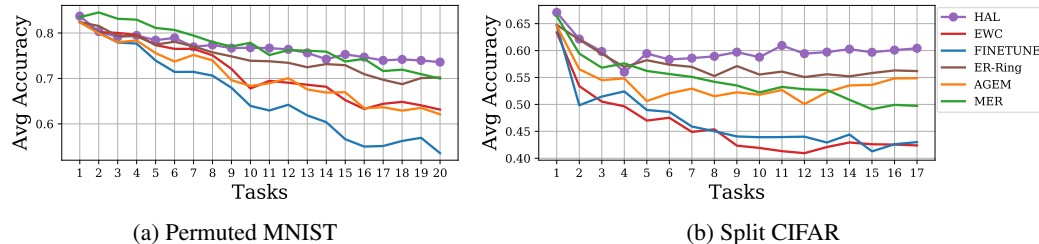

(a) Permuted MNIST           (b) Split CIFAR

Figure 1: Evolution of Accuracy (Eq. 2) as new tasks are learned. When used, episodic memories contain up to one example per class per task.

Table 2: Accuracy (Eq. 2) results for large (3 to 5 examples per class per task) episodic memory sizes. Here we only compare methods that use an episodic memory. Averages and standard deviations are computed over five random seeds.

| Method | Permuted MNIST | | Rotated MNIST | |
|---|---|---|---|---|
| | $\|\mathcal{M}\| = 600$ | $\|\mathcal{M}\| = 1000$ | $\|\mathcal{M}\| = 600$ | $\|\mathcal{M}\| = 1000$ |
| AGEM | 63.2 (±1.47) | 64.1 (±0.74) | 49.9 (±1.49) | 53.0 (±1.52) |
| MER | 74.9 (±0.49) | 78.3 (±0.19) | 76.5 (±0.30) | 77.3 (±1.13) |
| ER-Ring | 73.5 (±0.43) | 75.8 (±0.24) | 74.7 (±0.56) | 76.5 (±0.48) |
| **HAL (ours)** | **76.2** (±0.52) | **78.4** (±0.27) | **77.0** (±0.66) | **78.7** (±0.97) |

| Method | Split CIFAR | | Split miniImageNet | |
|---|---|---|---|---|
| | $\|\mathcal{M}\| = 255$ | $\|\mathcal{M}\| = 425$ | $\|\mathcal{M}\| = 255$ | $\|\mathcal{M}\| = 425$ |
| AGEM | 56.9 (±3.45) | 59.9 (±2.64) | 51.6 (±2.69) | 54.3 (±1.56) |
| MER | 57.7 (±2.59) | 60.6 (±2.09) | 49.4 (±3.43) | 54.8 (±1.79) |
| ER-Ring | 60.9 (±1.44) | 62.6 (±1.77) | 53.5 (±1.42) | 54.2 (±3.23) |
| **HAL (ours)** | **62.9** (±1.49) | **64.4** (±2.15) | **56.5** (±0.87) | **57.2** (±1.54) |

## 5 RELATED WORK

In continual learning (Ring, 1997), also called lifelong learning (Thrun, 1998), a learner addresses a *sequence* of changing tasks without storing the complete datasets of these tasks. This is in contrast to *multitask learning* (Caruana, 1997), where the learner assumes simultaneous access to data from all tasks. The main challenge in continual learning is to avoid catastrophic interference or forgetting (McCloskey & Cohen, 1989; McClelland et al., 1995; Goodfellow et al., 2013), that is, the learner forgetting previously acquired knowledge when learning new tasks. The state-of-the art methods in continual learning can be categorized into three classes.

First, *regularization approaches* discourage updating parameters important for past tasks (Kirkpatrick et al., 2016; Aljundi et al., 2018; Nguyen et al., 2018; Zenke et al., 2017). While efficient in terms of memory and computation, these approaches suffer from brittleness due to feature drift as the number of tasks increases (Titsias et al.,

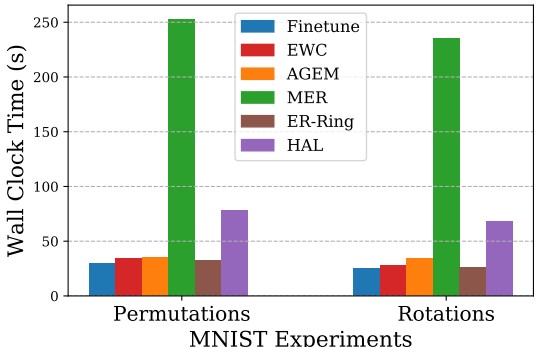

Figure 2: Training time (s) of MNIST experiments for the entire continual learning experience. MER and HAL both use meta-learning objectives to reduce forgetting.

Table 3: Impact of anchor selection, where we compare random-noise anchors (Random-), one random example-per-class anchors (Data-), and our optimized anchor selection (HAL).

| Anchor type | Permuted MNIST | | Split CIFAR | |
| --- | --- | --- | --- | --- |
| | Accuracy | Forgetting | Accuracy | Forgetting |
| ER-Ring | 70.2 | 0.12 | 56.2 | 0.13 |
| Random-Anchors | 72.7 | 0.10 | 57.8 | 0.13 |
| Data-Anchors | 73.2 | 0.10 | 59.0 | 0.12 |
| **HAL (ours)** | **73.6** | **0.09** | **60.4** | **0.10** |

2019). Additionally, these approaches are only effective when we can perform multiple passes over each dataset (Chaudhry et al., 2019a), a case deemed unrealistic in this work.

Second, *modular approaches* use different parts of the prediction function for each new task (Fernando et al., 2017; Aljundi et al., 2017; Rosenbaum et al., 2018; Chang et al., 2018; Xu & Zhu, 2018; Ferran Alet, 2018). Modular approaches do not scale to a large number of tasks, as they require searching over combinatorial space of module architectures. Another modular approach (Rusu et al., 2016; Lee et al., 2017) adds new parts to the prediction function as new tasks are learned. By construction, modular approaches have zero forgetting, but their memory requirements increase with the number of tasks.

Third, *episodic memory approaches* maintain and revisit a small episodic memory of datum from past tasks. In some of these methods (Li & Hoiem, 2016; Rebuffi et al., 2017), examples in the episodic memory are replayed and predictions are kept invariant by means of distillation (Hinton et al., 2014). In other approaches (Lopez-Paz & Ranzato, 2017; Chaudhry et al., 2019a; Aljundi et al., 2019b) the episodic memory is used as an optimization constraint that discourages increases in loss at past tasks. More recently, several works (Hayes et al., 2018; Riemer et al., 2019; Rolnick et al., 2018; Chaudhry et al., 2019b) have shown that directly optimizing the loss on the episodic memory, also known as experience replay, is cheaper than constraint-based approaches and improves prediction performance. Our contribution in this paper has been to improve *experience replay methods* with task anchors learned in hindsight.

There are other definitions of continual learning, such as the one of task-free continual learning. The task-free formulation does not consider the notion of tasks, and instead works on undivided data streams (Aljundi et al., 2019a,b). We have focused on the task-based definition of continual learning and, similar to many recent works (Lopez-Paz & Ranzato, 2017; Hayes et al., 2018; Riemer et al., 2019; Chaudhry et al., 2019a), assumed that only a *single pass through the data* was possible.

Finally, our gradient-based learning of anchors bears a similarity to (Simonyan et al., 2014) and (Wang et al., 2018). In Simonyan et al. (2014), the authors use gradient ascent on class scores to find saliency maps of a classification model. Contrary to them, our proposed hindsight learning objective optimizes for the forgetting metric, as reducing it is necessary while learning continually. Dataset distillation (Wang et al., 2018) proposes to encode the entire dataset in a few synthetic points at a given parameter vector by a gradient-based optimization process. Their method requires access to the entire dataset of a task for optimization purposes. We, instead, learn anchors in hindsight from the replay buffer of past tasks *after* training for current task. While Wang et al. (2018) aim to replicate the performance of the entire dataset from the synthetic points, we focus on reducing forgetting of an already learned task.

## 6    CONCLUSION

We introduced HAL, a meta-learning objective for continual learning. In our approach, we learn one "anchor point" per class per task, where predictions are requested to remain invariant. These anchors are learned using gradient-based optimization, and represent points that would maximize the forgetting of the current task throughout the entire learning experience. We simulate the forgetting that would happen during the learning of future tasks *in hindsight*, that is, by taking temporary gradient steps across a small episodic memory of past tasks. As shown in our experiments, anchoring

in hindsight complements and improves the performance of continual learning methods based on experience replay, achieving a new state of the art on four standard continual learning benchmarks.

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

APPENDIX

In Section A, we are report the forgetting 3 metric when large episodic memories are used. Section B provides the grid considered for hyper-parameters. Section C provides pseudo-code for HAL.

## A   MORE RESULTS

Table 4: Forgetting (Eq. 3) results for large (3 to 5 examples per class per task) episodic memory sizes. Here we only compare methods that use an episodic memory. Averages and standard deviations are computed over five random seeds.

| Method | Permuted MNIST | | Rotated MNIST | |
|---|---|---|---|---|
| | $|\mathcal{M}| = 600$ | $|\mathcal{M}| = 1000$ | $|\mathcal{M}| = 600$ | $|\mathcal{M}| = 1000$ |
| AGEM | 0.20 (±0.01) | 0.19 (±0.01) | 0.41 (±0.01) | 0.38 (±0.01) |
| MER | 0.14 (±0.01) | 0.09 (±0.01) | **0.12** (±0.01) | **0.11** (±0.01) |
| ER-Ring | 0.09 (±0.01) | 0.07 (±0.01) | 0.15 (±0.01) | 0.13 (±0.01) |
| **HAL (ours)** | **0.07** (±0.01) | **0.05** (±0.01) | **0.12** (±0.01) | **0.11** (±0.01) |

| Method | Split CIFAR | | Split miniImageNet | |
|---|---|---|---|---|
| | $|\mathcal{M}| = 255$ | $|\mathcal{M}| = 425$ | $|\mathcal{M}| = 255$ | $|\mathcal{M}| = 425$ |
| AGEM | 0.13 (±0.03) | 0.10 (±0.02) | 0.10 (±0.02) | 0.08 (±0.01) |
| MER | 0.11 (±0.01) | 0.09 (±0.02) | 0.12 (±0.02) | 0.07 (±0.01) |
| ER-Ring | 0.09 (±0.01) | **0.06** (±0.01) | 0.07 (±0.02) | 0.08 (±0.02) |
| **HAL (ours)** | **0.08** (±0.01) | **0.06** (±0.01) | **0.06** (±0.02) | **0.06** (±0.01) |

## B   HYPER-PARAMETER SELECTION

In this section, we report the hyper-parameters grid considered for experiments. The best values for different benchmarks are given in parenthesis.

- Multitask
  - learning rate: [0.003, 0.01, 0.03 (CIFAR, miniImageNet), 0.1 (MNIST perm, rot), 0.3, 1.0]
- Clone-and-finetune
  - learning rate: [0.003, 0.01, 0.03 (CIFAR, miniImageNet), 0.1 (MNIST perm, rot), 0.3, 1.0]
- Finetune
  - learning rate: [0.003, 0.01, 0.03 (CIFAR, miniImageNet), 0.1 (MNIST perm, rot), 0.3, 1.0]
- EWC
  - learning rate: [0.003, 0.01, 0.03 (CIFAR, miniImageNet), 0.1 (MNIST perm, rot), 0.3, 1.0]
  - regularization: [0.1, 1, 10 (MNIST perm, rot, CIFAR, miniImageNet), 100, 1000]
- AGEM
  - learning rate: [0.003, 0.01, 0.03 (CIFAR, miniImageNet), 0.1 (MNIST perm, rot), 0.3, 1.0]
- MER

- learning rate: [0.003, 0.01, 0.03 (MNIST, CIFAR, miniImageNet), 0.1, 0.3, 1.0]
- within batch meta-learning rate: [0.01, 0.03, 0.1 (MNIST, CIFAR, miniImageNet), 0.3, 1.0]
- current batch learning rate multiplier: [1, 2, 5 (CIFAR, miniImageNet), 10 (MNIST)]

- ER-Ring
  - learning rate: [0.003, 0.01, 0.03 (CIFAR, miniImageNet), 0.1 (MNIST perm, rot), 0.3, 1.0]

- HAL
  - learning rate: [0.003, 0.01, 0.03 (CIFAR, miniImageNet), 0.1 (MNIST perm, rot), 0.3, 1.0]
  - regularization ($\lambda$): [0.01, 0.03, 0.1 (MNIST perm, rot), 0.3 (miniImageNet), 1 (CIFAR), 3, 10]
  - mean embedding strength ($\gamma$): [0.01, 0.03, 0.1 (MNIST perm, rot, CIFAR, miniImageNet), 0.3, 1, 3, 10]

## C   HAL ALGORITHM

Algorithm 1 provides pseudo-code for HAL.

---

**Algorithm 1** Training of HAL on sequential data $\mathcal{D} = \{\mathcal{D}_1, \cdots, \mathcal{D}_T\}$, with total replay buffer size 'mem_sz', learning rate '$\alpha$', regularization strength '$\lambda$', mean embedding decay '$\beta$', mean embedding strength '$\eta$'.

---

1: **procedure** HAL($\mathcal{D}$, mem_sz, $\alpha, \lambda, \beta$)
2:    $\mathcal{M} \leftarrow \{\} * \text{mem\_sz}$
3:    $\{e_1, \cdots, e_T\} \leftarrow \{\}$
4:    **for** $t \in \{1, \cdots, T\}$ **do**
5:        $\phi_t \leftarrow \vec{0}$
6:        **for** $\mathcal{B} \sim \mathcal{D}_t$ **do**                                                        ▷ Sample a batch from current task
7:            $\mathcal{B}_\mathcal{M} \sim \mathcal{M}$                                                        ▷ Sample a batch from episodic memory
8:            $\tilde{\theta} \leftarrow \theta - \alpha \cdot \nabla_\theta \ell(\mathcal{B} \cup \mathcal{B}_\mathcal{M})$                                        ▷ Temporary parameter update
9:            $\theta \leftarrow \theta - \alpha \cdot \nabla_\theta \left( \ell(\mathcal{B} \cup \mathcal{B}_\mathcal{M}) + \lambda \cdot \sum_{t' < t} (f_\theta(e_{t'}, t') - f_{\tilde{\theta}}(e_{t'}, t'))^2 \right)$   ▷ Anchoring objective (Eq. 5)
10:           $\phi_t \leftarrow \beta \cdot \phi_t + (1 - \beta) \cdot \phi(\mathcal{B})$                              ▷ Running average of mean embedding
11:           $\mathcal{M} \leftarrow \text{UpdateMemory}(\mathcal{M}, \mathcal{B})$                              ▷ Add samples to a ring buffer
12:       **end for**
13:       $e_t, \theta \leftarrow \text{GetAnchors}(\mathcal{M}, \theta, \phi_t, \eta)$                          ▷ Get anchors for current task
14:    **end for**
15:    **return** $\theta, \mathcal{M}$
16: **end procedure**

 

1: **procedure** GETANCHORS($\mathcal{M}, \theta_t, \phi_t, \gamma$)
2:    $\theta \leftarrow \theta_t$
3:    **for** $\mathcal{B}_\mathcal{M} \sim \mathcal{M}$ **do**
4:        $\theta \leftarrow \theta - \alpha \cdot \nabla_\theta \ell(\mathcal{B}_\mathcal{M})$                          ▷ Finetune $\theta_t$ by taking SGD steps on the episodic memory
5:    **end for**
6:    $\theta_\mathcal{M} \leftarrow \theta$                                                        ▷ Store the updated parameter
7:    $e_t \leftarrow \text{rand}()$                                                        ▷ Initialize the task anchors
8:    **for** $1, \cdots, k$ **do**
9:        $e_t \leftarrow e_t + \alpha \cdot \nabla_{e_t} \left( \ell(f_{\theta_\mathcal{M}}(e_t, t), y_t) - \ell(f_{\theta_t}(e_t, t), y_t) - \gamma (\phi(e_t) - \phi_t)^2 \right)$   ▷ Maximize forgetting (Eq. 9)
10:   **end for**
11:   **return** $e_t, \theta_t$
12: **end procedure**

---

## D    APPROXIMATION OF ANCHORING GRADIENT

Here we derive the approximation of anchoring objective (Eq. 5) gradient. In particular, we are more interested in the regularization part of the anchoring objective that involves second-order terms. We refer to this gradient as $g_{anch}$. We follow similar arguments as (Nichol & Schulman, 2018).

Let $\theta_0$ be the parameter vector before the temporary update, $\ell_{ce}$ be the cross-entropy loss, and $\ell_{L2}$ be the L2 loss in the anchoring objective ( Eq. 5 in the main paper). We will use the following definitions:

$$
\begin{aligned}
\overline{g}_0 &= \ell'_{ce}(\theta_0) && \text{(gradient of cross-entropy loss at initial point on } \mathcal{B} \cup \mathcal{B}_{\mathcal{M}}) \\
\overline{H}_0 &= \ell''_{ce}(\theta_0) && \text{(Hessian of cross-entropy loss at initial point on } \mathcal{B} \cup \mathcal{B}_{\mathcal{M}}) \\
\overline{g}_1 &= \ell'_{L2}(\theta_0) && \text{(gradient of L2 loss at initial point on anchors)} \\
\overline{H}_1 &= \ell''_{L2}(\theta_0) && \text{(gradient of L2 loss at initial point on anchors)}
\end{aligned}
$$

Let $U_0 = \theta_0 - \alpha\overline{g}_0$ be the operator giving a temporary update in the two-step process of (Eq. 5), and let $\theta_1$ be the temporary update (note that $\tilde{\theta}$ is used in the main paper). The $g_{anch}$ is given by:

$$
\begin{aligned}
g_{anch} &= \frac{\partial}{\partial \theta_0} \ell_{L2}(U_0) \\
&= U'_0 \cdot \ell'_{L2}(\theta_1) \\
&= \left(I - \alpha\overline{H}_0\right) \cdot \ell'_{L2}(\theta_1)
\end{aligned}
\tag{10}
$$

Let's calculate the first order Taylor's approximation of $\ell'_{L2}(\theta_1)$

$$
\begin{aligned}
\ell'_{L2}(\theta_1) &= \ell'_{L2}(\theta_0) + \ell''_{L2}(\theta_0) \cdot (\theta_1 - \theta_0) + O(||\theta_1 - \theta_0||^2) \\
&= \overline{g}_1 + \overline{H}_1 \cdot (\theta_0 - \alpha\overline{g}_0 - \theta_0) + O(\alpha^2) \\
&= \overline{g}_1 - \alpha\overline{H}_1 \cdot \overline{g}_0 + O(\alpha^2)
\end{aligned}
\tag{11}
$$

Putting Eq. 11 in Eq. 10 and simplifying yields:

$$
g_{anch} = \overline{g}_1 - \alpha\overline{H}_1 \cdot \overline{g}_0 - \alpha\overline{H}_0 \cdot \overline{g}_1 + O(\alpha^2)
\tag{12}
$$

This form is very similar to second-order MAML gradient formulation, Eq. 25 in (Nichol & Schulman, 2018). Further simplification of terms like $(\overline{H}_1 \cdot \overline{g}_0)$ would yield inner product between the gradients $\overline{g}_0$ and $\overline{g}_1$. However, unlike MAML (Finn et al., 2017), Reptile (Nichol & Schulman, 2018) or MER (Riemer et al., 2019), in anchoring objective these gradients correspond to different losses.

