# OpenReview forum: "Using Hindsight to Anchor Past Knowledge in Continual Learning"
_ICLR.cc/2020/Conference — Reject_

### Official Review · AnonReviewer2 · 2019-10-19
**Official Blind Review #2**

**Rating:** 6

**Review:**

The technique of replay is well established. You have added a selection criterion for what to replay. You show a modest improvement over a random sample. You choose to view this as proof of the criterion, I on the other hand see it as proof of a very small result. To me it seems more like a negative result. I believe negative results are as important as positives. So I recommend acceptance.

You may want to expand the selection criterion you evaluate. You selected the most leverage on forgetting what happens as you select less and less leveraged cases? How strong is this effect? How linear? etc...

**Experience Assessment:**

I have read many papers in this area.

**Review Assessment: Checking Correctness Of Derivations And Theory:**

I assessed the sensibility of the derivations and theory.

**Review Assessment: Checking Correctness Of Experiments:**

I assessed the sensibility of the experiments.

**Review Assessment: Thoroughness In Paper Reading:**

I read the paper thoroughly.

---

> ### Author Response · Authors · 2019-11-09
> **To Reviewer 2**
>
> We thank the reviewer for providing feedback on the draft.
>
> The purpose of the work is not to study a selection criterion for populating replay buffer, which has been well-established as pointed out by the reviewer, but what more can be achieved from the replay buffer irrespective of the selection criterion. Towards this, we make two contributions; 1) we propose a meta-learning inspired anchoring objective (Eq. 5), and 2) we use replay buffer to synthesize anchors (which are different from examples stored in the replay buffer). Both these changes improve average accuracy by 5% and reduce forgetting by 20% over a standard experience replay baseline (assuming the same sample selection criterion).
>
> Since, in a continual learning setting, Forgetting is one of the main metrics that we want to reduce, we synthesize anchors that are optimized for this metric. To simulate forgetting we use replay buffer as a proxy to future and construct anchors that give maximum forgetting on this buffer. The other measure that can be leveraged is Accuracy but without the access to train or test set of the current task, it’s impossible to optimize over this measure.

---

### Official Review · AnonReviewer1 · 2019-10-24
**Official Blind Review #1**

**Rating:** 6

**Review:**

*** Summary
This paper proposes to formulate the continual learning problem as a meta-learning problem where the multi-tasking and forgetting can be addressed simutaneously. To assess the forgetting, this paper suggests using the anchor points learned in hindsight. Empirical results show that this formulation performs better than previous methods.

*** Strengths
1. The idea of using meta-learning to deal with the forgetting issue is interesting. Using anchor points to represent previous tasks is a sensible solution.

2. The comparison with the previous shows that the proposed method performances consistently well on multiple tasks.


*** Weakness
1. The results in Table 3 deserves more discussion. The differences between different anchor are marginal compared with the standard deviations in Table 1. Especially, the result of Data-Anchors and HAL can be treated as the same. I wonder if the authors have any intuition behind this. Basically, this result is not persuasive for the effectiveness of the proposed anchor point learning method. Also, I’m curious to see what would happen if you have oracle access to future tasks which means we don’t need to learn the anchor points in hindsight.

2. The writing can be polished. There are several typos. For example, in the third line of page 3, “and following“ might be “following”. In the first line of Section 3, the colon after “state of the art” should be removed.


**Experience Assessment:**

I have read many papers in this area.

**Review Assessment: Checking Correctness Of Derivations And Theory:**

I carefully checked the derivations and theory.

**Review Assessment: Checking Correctness Of Experiments:**

I carefully checked the experiments.

**Review Assessment: Thoroughness In Paper Reading:**

I read the paper thoroughly.

---

> ### Author Response · Authors · 2019-11-09
> **To Reviewer 1**
>
> We thank the reviewer for providing feedback on the draft. Below we answer questions asked by the reviewer:
>
> 1) The overall contribution of the paper is two folds; proposing a meta-learning inspired anchoring objective and providing a way to construct synthetic anchors in a streaming setting without access to current and future datasets. In the main tables (1, 2), we reported the best results which we got by combining anchoring objective with HAL anchors. Overall, our method improves accuracy by +5% and reduce forgetting by 20% over the best baseline. While using anchors does bring a statistically significant improvement over standard experience replay, in Tab. 3, we only compare different anchoring strategies. In this ablation, the proposed anchor point learning gives small but consistent improvements compared to when real data samples are used as anchors. We agree with the author, that for easy datasets, such as MNIST, the performance gain is not statistically significant, which we attribute to easy nature of MNIST benchmark, but for a more complex dataset and architecture (CIFAR & ResNet-18), the performance gain in average accuracy is +1.4 in absolute points which is higher than the standard deviation of +-0.52 in that experiment. Using only a single example in memory, we believe, this gain is significant. Upon the suggestion of the reviewer, below we report the results of an experiment that assumes oracle access to future datasets for constructing anchors.
>
>                                   MNIST                                                       CIFAR
> ----------------------------------------------------------------------------------------------------------------------
>                  Accuracy             Forgetting                    Accuracy             Forgetting
> ----------------------------------------------------------------------------------------------------------------------
> HAL       73.6 (+- 0.31)         0.09 (+- 0.01)                60.4 (+-0.54)       0.10 (+- 0.01)
> Oracle   73.9 (+- 0.41)         0.09 (+- 0.01)                61.1 (+- 0.94)      0.09 (+- 0.01)
>
> Note, having oracle access to future datasets defies the spirit of continual learning.
>
> 2) We have updated the draft. We are hopeful that it reads better now.

---

### Official Review · AnonReviewer4 · 2019-11-01
**Official Blind Review #4**

**Rating:** 6

**Review:**

This paper proposes an approach for continual learning that improves existing memory / replay -based methods, by learning anchors for each previous task. This is achieved by computing a temporary parameter update with a conventional loss (on both the current task and samples from an episodic memory buffer); and trading off this update with an additional loss measuring the reduction in performance on a set of "anchor samples" from previous tasks. Gradient descent is used to estimate the task-specific anchors that are most crucial across all (past and future) tasks.

The paper is well-written, it is a novel and well-explained idea, and the results are compelling. I lean towards acceptance, but I think there are a number of things that could be improved before final publication.

1) On how the anchors are estimated:
a) It is stated on page 3 that "... choose each e_t as a tool to minimize the Forgetting metric (Eq. 3). That is, we are interested in letting e_t be the example from task t that would maximize the amount of forgetting about task t throughout the entire learning experience...."
This appears to be contradictory - is the aim to find the example that would maximise the amount of forgetting for a given task after training on all tasks, in order to keep samples that are most "crucial"?
This section could be clearer, and further discussion and intuition would be beneficial for the reader.
b) Does it always make sense to choose anchors that maximise forgetting on the current task? Is this akin to finding examples that would optimally alter the decision boundary, in a similar fashion to support vectors?
c) I really like the intuition behind optimizing for the anchors via gradient descent (since finding examples that would be most forgotten in future violates the continual learning assumption). It would be interesting to visualize what anchors are found. For example, do they tend to be close to the centroid of a class (such that the mean embedding regulariser is low)? Are they examples that tend to be visually close to other classes, or are they outliers?
d) Given that anchors are already estimated and used to alleviate forgetting, what's the benefit of the additional episodic memory? An ablation is performed with different memory sizes, but what if *only* anchors are used (ie. a size of zero)? This would be useful in delineating the benefits of the anchors versus standard replay.

2) On the two-step optimisation (Eqn. 5):
a) It's not clear to me why this constitutes a meta-learning process: there is no adaptation at test time, nor does this perform task inference or learn how to learn. In this case, the method just uses a single gradient step to compute the change in predictions at the anchor points.
b) I wonder if there is a relationship to the loss used in Riemer et al, ICLR 2019. In particular, it seems that the gradient dot product term (used in that work to maximise transfer) may be related to a first-order Taylor expansion of the L2 term in this work (the change in predictions at anchor points, Eqn 5). It would be good to clarify whether this is the case, and add further discussion and intuition as needed.

3) On the relationship to existing work:
a) The approach seems conceptually most similar to iCARL (cited in the paper), which maintains "exemplars" from each previous task in order to avoid forgetting. I think the writing could be much more explicit about this, and both (i) compare against the performance of iCARL, and (ii) clearly discuss the differences between the iCARL exemplars and the anchors obtained in this work.
The equivalent result for splitCIFAR in the iCARL paper (for 20 tasks x 5 classes) appears to be around 45-50%, so I think it is necessary to compare and evaluate the source of this large improvement in performance.
b) Another paper worth discussing and comparing further is Variational Continual Learning - which is effectively a regularization-based method augmented with a coreset / episodic memory.
c) The paper seems quite centred on memory/replay- based approaches to continual learning; the abstract in particular says "SOTA continual learning methods implement different types of experience replay", but this fails to focus on non- memory-based methods. I would temper the claim to "many approaches to continual learning...", and spread the focus beyond replay-based approaches.

**Experience Assessment:**

I have published one or two papers in this area.

**Review Assessment: Checking Correctness Of Derivations And Theory:**

I assessed the sensibility of the derivations and theory.

**Review Assessment: Checking Correctness Of Experiments:**

I assessed the sensibility of the experiments.

**Review Assessment: Thoroughness In Paper Reading:**

I read the paper thoroughly.

---

> ### Author Response · Authors · 2019-11-09
> **To Reviewer 4 [1/2]**
>
> We thank the reviewer for providing thorough comments for improving the quality of the manuscript. Below we answer some of the questions asked by the reviewer.
>
> 1.a) For a task t, the idea is to find (or synthesize) an anchor where the model would have undergone a maximum change had training with future tasks (with t’>t) been performed. Then, keeping the model prediction intact on such an anchor would maximally preserve the performance of task t. The change in performance of a task is associated with an increase in loss on that task, which can be attributed to forgetting. Finding an anchor where the current task is forgotten the most, and subsequently training on it, would reduce the forgetting metric maximally. We have clarified this in the updated draft.
>
> 1.b) We believe that, given a tight memory budget, an anchor that maximally reduces forgetting will always be useful in making the model less forgetful. A model that is less forgetful would transfer more to future tasks. The exact interpretation in terms of decision boundary is tricky, but, as the reviewer pointed out, such anchors could lie very close to the decision boundary such that any subsequent change in the classifier during future training degrades their performance maximally.
>
> 1.c) In our visualizations, we could not find many semantic similarities between these anchors and classes.
>
> 1.d) The proposed synthetic anchors are complementary to the real examples in the episodic memory. While these anchors encode the information of task t and its corresponding trained parameter vector (theta_t), they do not encode semantic information of classes present in the task — which is captured by the episodic memory. Furthermore, episodic memory itself is used in learning these anchors. If we are keeping episodic memory to synthesize these anchors, we might use the memory for actual training purposes as well. In our experiments, training with memory always performs better than training without memory. Below we report numbers on Permuted MNIST experiment with and without episodic memory:
>
>                                                                            Accuracy                                      Forgetting
> --------------------------------------------------------------------------------------------------------------------
> Baseline (no memory or anchors)              53.5 (+-1.46)                               0.29 (+-0.01)
> Anchors                                                           62.5 (+- 1.12)                              0.20 (+- 0.01)
> Memory                                                           70.2 (+- 0.56)                              0.12 (+- 0.01)
> Memory + Anchors                                        73.6 (+- 0.31)                              0.09 (+- 0.01)
>
> As can be seen from the numbers, training with episodic memory significantly boosts the final performance.
>
> 2.a) We call it a meta-learning process owing to its similarity with gradient-based meta-learning approaches, such as MAML (Finn et al.), where nested optimization similar to Eq. 5 is used. However, we restrict the inner update to only a single gradient step. The reviewer is correct in pointing out that unlike “learning to learn” approaches there are no meta-variables to optimize over in our case. We have clarified it in the updated draft.
>
> 2.b) This is an excellent point and we thank the reviewer for pointing it out. We derive the formulation of the anchoring objective gradient and it indeed contains dot product gradient terms. The derivation is given in Appendix D of the updated draft. There are some differences, however.
>
>      a) Unlike MAML, Reptile, and MER (Riemer et al.), in the anchoring objective the dot product arises between gradients coming from different loss functions — cross-entropy loss in the inner update and L2 loss in the outer update.
>
>      b) The application of Reptile in MER assumes a stationary distribution and as such applying Reptile in a non-sequential setting is non-trivial. MER achieves this stationary distribution by Reservoir sampling of the episodic memory. Anchoring objective assumes no such condition. In fact, in our experiments, we use FIFO buffer as episodic memory and show significant improvement over MER (Riemer et al.).

---

> > ### Author Response · Authors · 2019-11-09
> > **To Reviewer 4 [2/2]**
> >
> > 3.a) There are some important differences with iCaRL.
> >
> >     a) In iCaRL, the exemplars are the *actual* data samples that lie closest to the average feature vectors, whereas we learn synthetic anchors by means of gradient-based optimization.
> >
> >     b) The numbers reported in the original iCaRL paper are for multiple-epochs setup, whereas we are interested in the ‘single-pass through the data’ protocol as studied by many recent works.
> >
> >     c) In their work A-GEM, Chaudhry et al. 2018 [1], compared iCaRL with other recent memory-based methods, such as A-GEM, in a single-epoch setup, clearly showing significant improvements over iCaRL. In this work, we show significant improvement over A-GEM and other recent memory-based methods for the same size of episodic memory.
> >
> >   d) We ran iCaRL in the same setup as ours; single-epoch training and using an episodic memory of 1 exemplar per class per task, on Split-CIFAR benchmark. We used the nearest-mean-of-exemplars rule for classification as suggested in the iCaRL paper.  Following are the numbers:
> >
> >                                Accuracy (%)                                      Forgetting
> > -------------------------------------------------------------------------------------------
> > iCaRL                      46.4 (+-1.21)                                0.16 (+- 0.01)
> > ER-Ring                  56.2 (+-1.93)                                0.13 (+- 0.01)
> > HAL (Ours)            60.4 (+-0.54)                                0.10 (+- 0.01)
> >
> > For bigger memory setups we refer to the study of Chaudhry et al. [1]. It can be seen from the table that we significantly improve over iCaRL both in terms of accuracy and forgetting metric. The main source of improvement, we believe, is direct training over the memory, experience replay (ER-Ring), instead of using memory in the knowledge distillation loss. The anchoring objective further improves numbers quite significantly.
> >
> >
> > 3.b) We ran VCL with the same setup as ours; single-epoch training and using an episodic memory of 1 example per class per task, on Permuted MNIST benchmark. Below we report the results of the experiment:
> >
> >                                                          Accuracy (%)
> > ----------------------------------------------------------------------
> > VCL                                                      53.7
> > VCL + Random Coreset                    54.5
> > VCL + K-center Coreset                    56.3
> > HAL (Ours)                                         73.6
> >
> > Like other regularization-based continual learning methods, such as EWC, VCL also suffers a performance loss in the case of a single-epoch training. For VCL, we suspect this performance loss is due to noisy posterior approximation in the single-epoch setup.
> >
> > 3.c) We have updated the draft to reflect the reviewer’s suggestions.
> >
> >
> > ------------------------------------------------------------------------------------------------------------
> > [1] Efficient Lifelong Learning with A-GEM, Chaudhry et al., ICLR 2019

---

### Comment · AnonReviewer2 · 2019-10-18
**table 3**

Hi, in table 3 we see that the selected anchors vs randomly selected case makes almost zero difference. This seems like a worthwhile result in showing the current selection mechanism is not useful.

You need to proof read for grammar, particularly between eq 4 and 5.

---

> ### Author Response · Authors · 2019-10-19
> **HAL improves upon real Data-based anchors.**
>
> Thank you for your comment. If we understood correctly, by 'selected anchors' the reviewer means anchors learned in hindsight by HAL, and randomly selected anchors are 'Data-Anchors' containing *actual samples* from the tasks dataset. Table 3 shows that our proposed hindsight learned anchors, HAL, improve upon the real data-based anchors, Data-Anchors, in terms of average accuracy by +0.4 and +1.4 in absolute points on MNIST and CIFAR, respectively. Note, a similar improvement in the Forgetting metric. The performance gain is nonzero and is more pronounced in a bigger network and datasets  (ResNet-18, CIFAR).
>
> We will fix the grammar.

---

### Author Response · Authors · 2019-11-09
**Paper Revision**

We thank all the reviewers for their positive reviews. We have uploaded a revision of the paper which accommodates all the reviewers' suggestions.

Below is the summary of the changes:

* Added the derivation of the gradient form of the anchoring objective.
* Updated Sec. 3 to explain the connection with meta-learning and improved the exposition of anchor learning.
* Miscellaneous fixes in the grammar.

We would welcome further suggestions, if any, from the reviewers to improve the quality of the manuscript.

---

### Author Response · Authors · 2020-01-12
**Unsatisfied with the paper decision**

We, respectfully, would like to mention our disagreement with the PCs regarding the grounds for the rejection of the paper. We do welcome fair criticism, however, unfortunately the final decision is not in line with the reviews we received. Despite getting three weak accepts, the paper has been rejected on grounds some of which are not even mentioned in any of the reviews. The final points which were mentioned as weaknesses of the paper could be rebutted had they been brought up during the rebuttal period. It is extremely discouraging that we weren't given any chance to defend our work, and the lack of proper discussion left us rather disappointed. Below we try to rebut the points made by the program committee in their final decision:

 * Grammatical Issues: To the best of our understanding, there were only two instances of grammatical issues in the original manuscript that were fixed in the draft we uploaded during the rebuttal period.

 * Experimental Details: Random ordering refers to different dataset orderings. This has been a standard practice for supervised continual learning benchmarks. We clearly mention this in our introduction and experiment sections.

 * Visualising anchor points: We had mentioned that the anchor points do not correspond to any semantic information in the class. A potential explanation could be that they are learned by maximizing forgetting, an objective that doesn't necessarily encode semantic information. Their importance lie in how they encode the task optimum and overall behavior of the classifier for that task. We can add the visualizations in the updated draft but, as mentioned, they don’t carry any semantically meaningful information.

 * NLP and RL experiments: We clearly mention in the introduction that we only test on supervised classification benchmarks on vision datasets — which has been standard practice in many CL works accepted to ICLR and NeurIPS. In fact, many CL papers accepted to ICLR 2020 only run experiments on MNIST and CIFAR. In addition to these two datasets, we also show superior performance on mini-ImageNet. We could have run experiments on RL tasks if it were mentioned during the review cycle.

 * Missing Citations: Would be great to know which papers we missed? We have no clues about that yet.

---

### Decision · Program_Chairs · 2019-12-19

**Decision:**

Reject

**Comment:**

This paper proposes a continual learning method that uses anchor points for experience replay. Anchor points are learned with gradient-based optimization to maximize forgetting on the current task. Experiments MNIST, CIFAR, and miniImageNet show the benefit of the proposed approach.

As noted by other reviewers, there are some grammatical issues with the paper.

It is missing some important details in the experiments. It is unclear to me whether the five random seeds how the datasets (tasks) are ordered in the experiments. Do the five random seeds correspond to five different dataset orderings? I think it would also be very interesting to see the anchor points that are chosen in practice. This issue is brought up by R4, and the authors responded that anchor points do not correspond to classes. Since the main idea of this paper is based on anchor points, it would be nice to analyze further to get a better understanding what they represent.

Finally, the authors only evaluate their method on image classification. While I believe the technique can be applied in other domains (e.g., reinforcement learning, natural language processing) with some modifications, without providing concrete empirical evidence in the paper, the authors need to clearly state that their proposed method is only evaluated on image classification and not sell it as a general method (yet).

The authors also miss citations to some prior work on memory-based parameter adaptation and its variants.

Regardless all of the above issues, this is still a borderline paper. However, due to space constraint, I recommend to reject this paper for ICLR.